

# Turbidivision: a machine vision application for estimating turbidity from underwater images

Ian M. Rudy[1] and Matthew J. Wilson[2]

[1] Department of Math and Computer Science, Susquehanna University, Sellinsgrove, Pennsylvania, United States
[2] Freshwater Research Institute, Susquehanna University, Sellinsgrove, Pennsylvania, United States

## ABSTRACT

The measurement of turbidity serves as a key indicator of water quality and purity, crucial for informing decisions related to industrial, ecological, and public health applications. As existing processes require both additional expenses and steps to be taken during data collection relative to photography, we seek to generate accurate estimations of turbidity from underwater images. Such a process could give new insight to historical image datasets and provide an alternative to measuring turbidity when lower accuracy is acceptable, such as in citizen science and education applications. We used a two-step approach to a machine vision model, creating an image classification model trained on image data and their corresponding turbidity values recorded from a turbidimeter that is then used to generate continuous values through multiple linear regression. To create a robust model, we collected data for model training from a combination of *in situ* field sites and lab mesocosms across suspended sediment and colorimetric profiles, with and without a Secchi disk for visual standard, and binned images into 11 classes 0–55 Formazin Nephelometric Units (FNU). Our resulting classification model is highly accurate with 100% of predictions within one class of the expected class, and 84% of predictions matching the expected class. Regression results provide a continuous value that is accurate to ±0.7 FNU of true values below 2.5 FNU and ±33% between 2.5 and 55 FNU; values that are less accurate than conventional turbidimeters but comparable to field-based test kits frequently used in classroom and citizen science applications. To make the model widely accessible, we have implemented it as a free and open-source user-friendly web, computer, and Google Play application that enables anyone with a modern device to make use of the tool, the model, or our repository of training images for data collection or future model development.

# INTRODUCTION

In fields ranging from environmental science to public health, assessing water quality is vital. These assessments of water quality often begin with measuring turbidity, which can impact water clarity (*Davies-Colley & Smith, 2001*) and potability

Corresponding author
Matthew J. Wilson,
wilsonmatt@susqu.edu

(*LeChevallier, Evans & Seidler, 1981*). Turbidity is an optical measure of water clarity, measured by the scattering of light by particles suspended in water, contributing to a murky or cloudy appearance. In nature, many of these particles are agitated sediment, such as clays and soils or suspended organic matter (*e.g.*, plant debris or microscopic organisms) and largely depend on surrounding land use (*Moreno Madriñán et al., 2012*). Pollutants contained in industrial and agricultural runoff can also be linked to turbidity (*Rügner et al., 2013*; *World Health Organization, 2017*).

Increasing turbidity can have negative impacts on aquatic life, since it often shares an inverse correlation with dissolved oxygen levels, creating an inhospitable environment for many taxa (*Talke, de Swart & De Jonge, 2009*). In addition, increasing turbidity can cause nonlethal effects on aquatic taxa through altered predator-prey interactions (*Abrahams & Kattenfeld, 1997*; *Ferrari, Lysak & Chivers, 2010*), decreasing light penetration and reduced photosynthesis (*Moore, Wetzel & Orth, 1997*), and physical stress such as gill deformities (*Lowe, Morrison & Taylor, 2015*). Beyond affecting aquatic ecosystems and taxa, overly turbid water is unsuitable for consumption by humans (*Muoio et al., 2020*) and livestock (*Umar et al., 2014*), as well as acting as an indicator of bacterial contamination (*Gharibi et al., 2012*). For water treatment plants, this necessitates filtering suspended particles to bring turbidity within acceptable levels of 5 Nephelometric Turbidity Units (NTU) or lower, depending on use and filtration method (*World Health Organization, 2017*). Furthermore, monitoring turbidity can be helpful in tracking sediment runoff and pollution in bodies of water, providing valuable insights for environmental management and conservation efforts (*Owens et al., 2005*).

Typically, turbidity is measured with a turbidimeter which shines a source of light–either white or infrared–into the fluid and a probe which measures the resulting scatter of light. Meters that measure in Formazin Nephelometric Units (FNU) shine an infrared light into the solution and measure the scatter at a 90-degree angle of incidence. Other common units for turbidity include NTU, which are measured with white light at a 90-degree angle of incidence, and Formazin Attenuation Units (FAU), which are measured with infrared light at a 180-degree angle of incidence (*Anderson, 2005*). Agency standards are typically based on method of measurement rather than accuracy thresholds. For example, U.S. Environmental Protection Agency standards require 90-degree hatchure and visible radiation, with equipment tested by the U.S. Geological Survey (USGS) having 5% error or less. These results are similar to the International Standards Organization (ISO) 7027 which require a back-scatter angle of 90 degrees. Testing by USGS found these instruments to also have less than 5% error at turbidity above 40 NTU and greater than 10% below 20 NTU (*Wilde & Gibs, 2008*).

Turbidimeters are generally expensive, costing hundreds or even thousands of dollars. For example, the LaMotte 2020i turbidimeter used during this project has an MSRP of $1,449 USD (https://lamotte.com/2020i-portable-turbidity-meter#specifications; LaMotte Company, Chesterton, MD, USA). This expense can be a significant barrier to citizen science initiatives, or laboratories with a limited budget, decreasing the ability of laypeople to contribute to water quality monitoring efforts. The converse can also be true, that lowering costs and creating easy and consistent methods of data collection increase the

quality of data and participation in data collection (*Zheng et al., 2018*; *Lee, Lee & Bell, 2020*). Measuring turbidity with a turbidimeter can also be time-consuming, as they necessitate the transport of the meter itself, or the collection of individual water samples for later measurement.

With the expansion and increasing accessibility of machine vision models their use for water quality has become an emerging field of research and offers potential to reduce time, costs, and increase accessibility of data collection. Current approaches include machine vision in conjunction with existing analytical tools (*Yan et al., 2024*), model development and image analysis from controlled environments (*Nazemi Ashani et al., 2024*), and remote sensing (*Leeuw & Boss, 2018*). The focus of machine vision research for water quality has largely been in economically important fields such as aquaculture (*Li & Du, 2022*) and wastewater treatment (*Mullins et al., 2018*). However, there has been little work with model development from *in situ* images for specific water quality parameters, such as turbidity. Furthermore, while historical underwater image datasets exist (*e.g.*, *Peng, Zhu & Bian, 2023*) we have been unable to identify any historical datasets of underwater images that include associated turbidity measurements, making it impossible to retroactively assess water quality for historical datasets.

Our primary goal was to develop a machine vision model capable of estimating turbidity from underwater images that could be made publicly available and easily accessible. Our model offers a cost-effective alternative to traditional turbidimeters, making water quality monitoring more accessible for those who may not have the necessary funds to purchase expensive equipment. Given the affordability and widespread availability of waterproof digital cameras, including smartphones, this approach has the potential to democratize non-critical water quality assessment (*e.g.*, *Zheng et al., 2022*). Using images can simplify the process for field sampling by requiring less equipment and eliminate sample processing. It also allows existing underwater image data without turbidity readings to be retroactively analyzed. We also evaluate the feasibility and accuracy of machine vision for predicting turbidity levels in water bodies. By assessing how well the model performs, this initiative could pave the way for innovative applications in research and citizen science, as well as gaining insights into historical data, fostering greater engagement and participation in environmental monitoring and conservation efforts.

## MATERIALS AND METHODS

### Photo/data collection

To develop this model, we paired two components in data collection: an underwater photograph and a measurement of the turbidity of the water source in the image. The turbidity was measured in FNU, and the readings were taken with a LaMotte 2020i Turbidity Meter. Two cameras were used for photography, an Olympus TG-6, and a Sony HDR-AS30V, with photos taken in the "Auto" photo mode. For each photo we recorded associated metadata for location, whether the water was flowing or not (y/n), whether a Secchi disk was present in the photo (y/n), camera information such as ISO, shutter speed, focal length, F-stop, white balance (when available), and the substrate present in the image (when possible). All photos were taken under ambient light conditions.

We collected field photos to represent a diversity of habitats with the intent of improving model robustness from water sources including rivers, lakes, ponds, and the ocean at turbidity levels between 0 and 55 FNU and calibrated the turbidity meter at each sampling location. The images were collected from bodies of water in Pennsylvania and Maryland, USA. Of the field photos taken, 298 photos were from lotic, 30 from lentic, 87 from brackish, and 25 from marine sources. We fixed the camera to the end of a metal rod and submerged it underwater either with or without a 4.2 cm Secchi disk in frame at either 12 or 23 cm from the camera. Images were collected by taking individual photographs while the camera was submerged and by saving a selection of frames from a video taken while the camera was submerged. We took care to avoid disturbing sediment to ensure the FNU reading that was recorded matched the FNU of the water in the image. We took controlled experimental photos in two different systems following the same procedure as field photos, an opaque 20 L bucket to compare with lentic field photos and a 100 L acrylic fish tank with an additional sheet of acrylic placed vertically in the tank center. The 100 L tank included pumps to create a recirculating system to compare with lotic photos from the field.

The standard procedure for data collected from the bucket followed a graduated increase in turbidity with underwater photos taken after each sediment addition. We added sediment in 4-g increments and stirred to homogenize. When the water settled, we would take another turbidity reading, and additional photos. We repeated this process multiple times until the turbidity reached the upper limit of our defined range (55 FNU). To increase the robustness of the model against colorimetric changes, we also collected photos with colored ink (J. Herbin: Perle Noire, Rouge Hematit, Gris Nuage, and Vert Empire) added to the water in increments of 0.01 mL/L, both with and without sediment. We repeated the same process in the recirculating aquarium with gravel added to the bottom of the tank for background texture. In total, we used 675 images in model development: 440 from the field, and 235 from the lab (114 from the bucket and 121 from the aquarium). For 38% of all photos we included the 4.2 cm diameter Secchi. Secchi and natural images were collected and equal proportions until training demonstrated no effect of Secchi presence on model effectiveness.

## Data preparation

Using the master image dataset, the images were broken into 11 groups (classes) based on the corresponding FNU values: 0–0.49, 0.5–0.99, 1–2.49, 2.5–4.99, 5–9.99, 10–14.99, 15–20.99, 21–28.99, 29–36.99, 37–44.99, 45–55. These classes were used as they represent the maximum number of classes we could create without impacting accuracy. We split data into training, testing, and validation groups without duplicate images within any two or all three of them. Approximately 75% of the images were used in training, 15% for testing, and 10% for validation. This training/testing/validation split follows the standard folder structure for classification datasets required for model training. All python scripts made for the project were run using Jupyter Lab, and the notebook files that contain them are included with the source code. The master dataset, including all images and associated data can be found in the GitHub repository or dataset archive (*Rudy & Wilson, 2024a*).

## Model training

We used the most recent release of You Only Look Once (YOLO), Ultralytics YOLOv8 model for training. The YOLO family of machine vision models includes object detection, image segmentation models, and classification models. We chose the YOLOv8-Classification model because of the well-established performance, computing times, and open access nature (*Jocher, Chaurasia & Qiu, 2023*; *Stormsson, Jocher & Xin, 2024*). YOLOv8 provides five different size models to use as the basis for transfer learning when training a model. From smallest to largest they include, Nano, Small, Medium, Large, and Extra Large. There are different sets of these models for object detection, instance segmentation, pose estimation, object detection with oriented bounding boxes, and classification tasks, all pretrained on the ImageNet dataset. We used the Large classification model as the starting point for transfer learning which uses the COCO dataset (*Lin et al., 2014*). Using transfer learning and starting with a pretrained model allowed us to increase the speed of the training process by eliminating the need to build up weights suited for classification, opting instead to modify a set of pre-existing weights to better suit the new task of estimating turbidity. The final model was trained to use $320 \times 320$ pixel images, resized from the dataset, and was trained for 15 epochs, with a batch size of 16. The final model was chosen as the model with the highest fitness score from a group of 50 models trained with the specified parameters, each with different hyperparameters mutated using the AdamW optimizer and called using the tuning function provided by Ultralytics (*Jocher, Chaurasia & Qiu, 2023*). The hyperparameters of the final model are provided alongside the archived source code (*Rudy & Wilson, 2024b*). Training all 50 models during the final tuning process took approximately 2.2 h using an Nvidia RTX 4080 model GPU.

## Regression prediction

Once the final classification model was trained, we performed inference on each image in the master dataset, which saves the confidence values between 0 and 1 of the top five most confident classes for each image in individual text files. From this we applied the confidence scores for each image and set all blank entries to a value of 0. We then compared these values to measured FNU by fitting a multiple linear regression model using the Weighted Least Squares method, with the predicted classes as the independent variables, and the measured value as the target variable to create a continuous estimate of turbidity in addition to the defined bins.

## RESULTS

The YOLOv8 training process generated a confusion matrix to visualize the performance of the model when inference was run on the validation set. An ideal model would have a perfect correlation, where each predicted class matches the actual class. For our model, predictions from the validation set were tightly clustered around the true classes, with 100% of predictions within one class of the expected class, and 84% of predictions matching the expected class (Fig. 1).

By comparison, the regression had an $R^2$ of 0.975, with the parameters ($x$) and coefficients ($\beta$) of the regression equation of form $y = \beta_0 + \beta_1 x_1 + \ldots + \beta_{11} x_{11}$ shown in

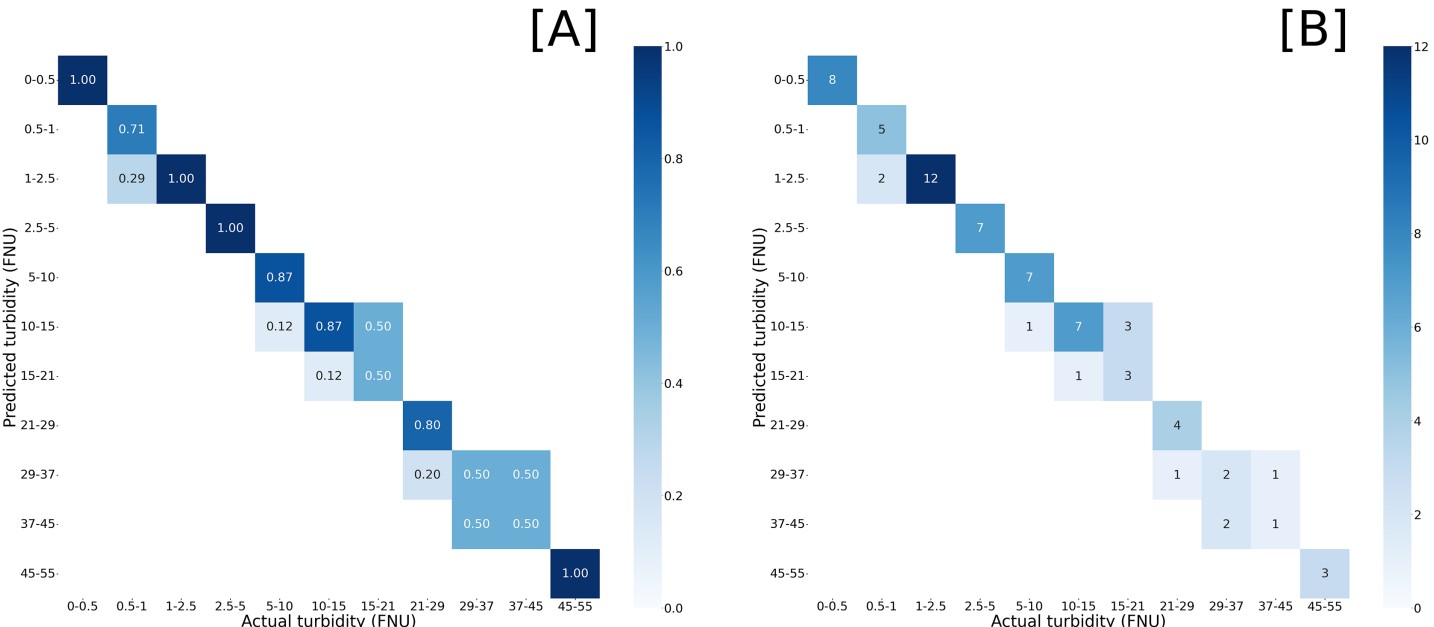

**Figure 1 Confusion matrices of model inference compared against validation set.** Predicted turbidity by (A) normalized values and (B) prediction counts against actual turbidity readings from *in situ* measurements. Darker blues represent closer match with proportional match or counts given within the heatmap.

Table 1, with additional regression metrics in Table 2. While the accuracy of the regression model is not significantly greater than the classification model, the numerical, rather than categorical, estimates provide a more broadly useful result for further analysis (Fig. 2). Given that our model predicts turbidity values in a range from 0 FNU to 55 FNU, assessing its accuracy based on the root mean square error (RMSE) does not make sense, as an error of approximately 2 FNU at the high end of the range is considerably less impactful than an error of 2 FNU at the low end. This is why relative root mean square error (RRMSE) and relative standard deviation (RSD) were calculated. While there is no accepted rule for RRMSE or RSD, the lower it is the more accurate the model is, and our 18.46% could generally be considered good. We found that from 0 to 2.5 FNU, 95% of our model's predictions fell within ±0.7 FNU of the true values, and that from 2.5 to 55 FNU, 95% of our model's predictions fell within ±33% of the true values. In addition, there was no change in model accuracy when trained based on metadata, such as water body type or Secchi disk presence allowing us to use all images together in the final dataset. We also visually compared subsets of the predicted and actual values against image appearance to confirm there were no patterns between image background and accuracy (Fig. 3).

## DISCUSSION

In this study, we evaluated the ability of a machine vision model to estimate turbidity and the accuracy of that model. While the accuracy of the model is lower than a physical meter, it is comparable to field test kits typically used in citizen science programs, free, and a more accessible way to measure turbidity for those willing to accept the reduced accuracy compared to a traditional turbidimeter as anyone with a smartphone can use the Android

**Table 1 Coefficients, 95% confidence interval, and significance of multiple linear regression by FNU class.** The lower and upper intervals (0.025 and 0.975, respectively) represent the upper and lower bounds of the 95% confidence interval for each FNU class. All *p* values were below the measurable limit and represented with 0 by the regression output.

| FNU class | Coefficient | Standard error | Lower interval (0.025) | Upper interval (0.975) | t | P > \|t\| |
|---|---|---|---|---|---|---|
| Intercept | −60.90 | 11.61 | −83.69 | −38.11 | −5.25 | 0 |
| 00–0.49 | 61.34 | 11.66 | 38.45 | 84.24 | 5.26 | 0 |
| 00.5–0.99 | 61.57 | 11.61 | 38.77 | 84.37 | 5.30 | 0 |
| 01–2.49 | 62.48 | 11.61 | 39.69 | 85.28 | 5.38 | 0 |
| 02.5–4.99 | 65.53 | 11.62 | 42.72 | 88.35 | 5.64 | 0 |
| 05–9.99 | 67.76 | 11.62 | 44.95 | 90.58 | 5.83 | 0 |
| 10–14.99 | 73.91 | 11.63 | 51.08 | 96.74 | 6.36 | 0 |
| 15–20.99 | 77.63 | 11.63 | 54.78 | 100.48 | 6.67 | 0 |
| 21–28.99 | 85.64 | 11.66 | 62.75 | 108.53 | 7.35 | 0 |
| 29–36.99 | 93.29 | 11.66 | 70.40 | 116.17 | 8.00 | 0 |
| 37–44.99 | 99.40 | 11.60 | 76.61 | 122.19 | 8.57 | 0 |
| 45–55 | 114.32 | 11.76 | 91.24 | 137.41 | 9.72 | 0 |

**Table 2 Statistical summary of the multiple linear regression model for predicted *vs.* actual turbidity.** Summary abbreviations include root mean square error (RMSE), relative root mean square error (RRMSE), and degrees of freedom (DF).

| Statistic | Value |
|---|---|
| Adj. R-squared | 0.975 |
| RMSE (standard deviation) | 2.055 |
| RRMSE (relative standard deviation) | 18.46% |
| F-statistic | 2,349 |
| Log-likelihood | −1,416.2 |
| AIC | 2,856 |
| Skew | 0.432 |
| Kurtosis | 11.997 |
| Number observations | 662 |
| DF residuals | 650 |
| DF model | 11 |

or web app. In addition, this model is accurate enough to provide insights on turbidity conditions from historical data-underwater image datasets that never had turbidity recorded alongside the images-which would otherwise be impossible to go back and obtain. As an open-source program, the model can also be integrated into other systems which could be used for any variety of other potential data processing tasks related to underwater imagery. Even though the linear regression model is no more accurate than the classification, a single numerical value with a known confidence level is often more useful for analysis.

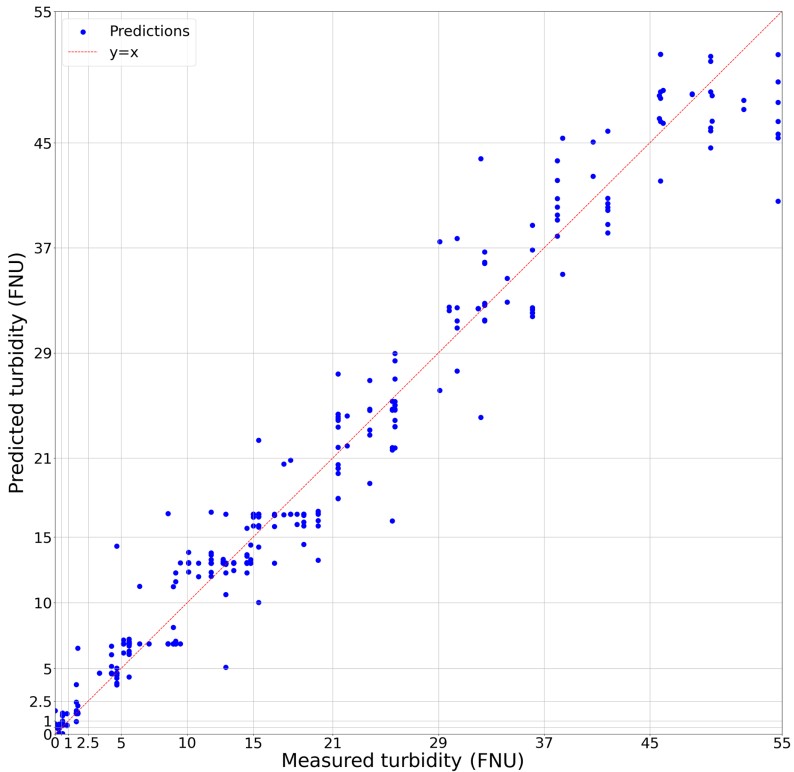

**Figure 2 Regression model of predicted turbidity against *in situ* turbidity.** The dashed red line represents x = y where predicted values would be a perfect match for *in situ* measurements. Each blue dot is a prediction of the linear regression based on the classification model confidence values of each image from the dataset. Grid lines represent classification bins used for image analysis.

## Web app

The Turbidivision web app combines the classification and regression models into one web-based program. It can run on any modern web browser, including those on desktop/laptop computers, as well as on phones and other mobile devices. Once the classification model is downloaded, the code runs fully on the client machine; the server is only used to provide the initial download of webpage code and assets, which should remain cached in the browser for future use. After processing each image, the model outputs confidence values for each class, and the linear regression converts that to a discrete numerical estimate. For each image processed, the name of the image, confidence values for each class, and output of the linear regression are added to a CSV file. The web app has a skeuomorphic GUI designed to look like a turbidimeter, but a version using basic unadorned html elements is available for increased compatibility. The user simply needs to click on the file input button, which opens the file browser, and select the image(s) they want to process. The images will then be processed on the user's computer, and a button to save the output CSV file will appear once the processing is complete. Binary distributions of the web app for Windows (.exe), Linux (AppImage), and Android (.apk), as well as the files for the web app for serving locally, are available (*Rudy & Wilson, 2024c*). In addition,

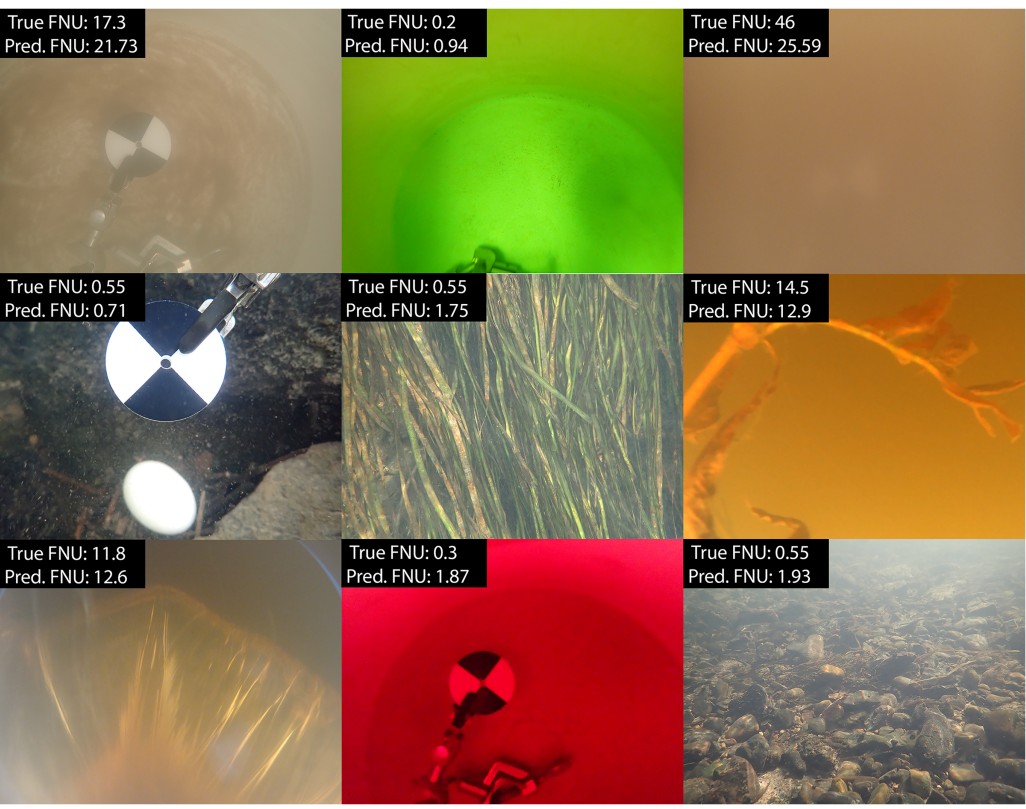

**Figure 3 Sample of predicted and measured turbidity values from field and lab images.** Images selected represent the diversity of locations, Secchi presence, substrate, angle, and dyes included in image collection and model training.     

the Android application has been released on the Google Play store as "Turbidivision" and the web app is available for use (*Rudy & Wilson, 2024d*).

## FUTURE DIRECTIONS

One potential improvement for future research could be changing the underlying model. We are predicting a continuous variable (turbidity), but instead of directly predicting a continuous variable, we first predicted a discrete variable, a class, and then used that to run a secondary regression model. If we were instead to create a neural network that ends in a fully connected layer which outputs a single continuous variable, from 0 to 55, we could effectively run regression directly on the images. However, such a process would require much more data and the creation of a custom convolutional neural network. The exact amount of data needed to train such a model would depend upon the architecture of the model itself, along with the desired accuracy. In addition, since regression is continuous, and not based on classification, direct model comparison is not possible. However, if we assume that quantizing turbidimeter measurements based on the reported accuracy of the meter provides an adequate analogue to number of classes, we can then calculate a minimum recommended amount of images needed for training such a regression model.

If we consider a goal to match the accuracy of the LaMotte 2020i turbidimeter, which is among the most accurate available (meters range from 2–5%; *Wilde & Gibs, 2008*), then the ideal would be within ± 0.05 FNU between 0–2.5 FNU (LaMotte Company, Chesterton, MD, USA). For this we can say that there are approximately 25 unique ranges of distinct accuracy within this range (2.5/0.1); and since it has a reported accuracy of ± 2% from 2.5–100 FNU, we can split it into 79 unique ranges of distinct accuracy between 2.5 and 55 FNU ($2.5 \times 1.04^{count} > 55$, *solved for count*). The sum of these two values, 104, gives us a rudimentary analogue to the number of classes needed. If we assume such a regression model would require a similar amount of data as a classification model and a recommended minimum of 150 images per class (*Shahinfar, Meek & Falzon, 2020*), we can use our analogue class count multiplied by 150 pictures per class to estimate an absolute minimum of 15,600 images, ideally with the turbidity values of these images evenly distributed between 0 and 55 FNU. Such a model would not be able to make use of the development speed increases of transfer learning, as all weights would need to be trained from scratch. This training would require a larger dataset, potentially larger than mentioned above, as well as more time required for model training. If such a model were developed with a similar architecture to Yolov8, processing times for the end user should be comparable to the model created in this project.

Within the existing modelling framework, more images, but fewer than those needed for a custom convolutional neural network, would improve the training process and should be able to achieve greater accuracy. More images would allow the model to learn the features of each class better, and a greater variety of images, such as in location, water quality, and lighting conditions, should help the model gain more resilience against features not related to turbidity. A larger dataset could also allow for the creation of more classes (smaller bins), which would allow the model to make more granular predictions. Another potential approach would be to train our model (or a similar model) to above-water images and assess accuracy. If a model could be trained on above-water images, even if more training is required, it would increase accessibility and the applicability to historical images dramatically.

## CONCLUSIONS

Measuring turbidity can be critically important for contexts ranging from human health to food webs. Our goal was to determine efficacy, accuracy, and precision of a machine vision model for measuring turbidity from underwater images. We successfully demonstrated the potential of modern machine vision techniques as viable for estimating the turbidity of natural bodies of water, with an accuracy comparable to commercial test kits up to 55 FNU and down to near 0 FNU (*e.g.*, LaMotte Turbidity Test Kit #7519-01). The model we created offers an accessible alternative to traditional turbidimeters and can provide turbidity measurements within an acceptable margin of error in many applications.

The application developed as part of this research project is also the first photo-based turbidity measuring tool accessible to the public. To make the model widely accessible, we implemented it as a free, user-friendly web application that would enable anyone with a

modern web-enabled device to make use of the tool. Additionally, the web application is designed to allow the data to be processed on the user's device, keeping their data secure and avoiding unnecessary use of internet bandwidth by preventing the need to upload their files to a server. The app is compatible with a wide range of devices and has a simple user-interface, allowing anyone to easily benefit from the results of this research. In addition, its use of images as an input allows users to retroactively gain insights on turbidity from historical underwater image datasets and understand past trends.

## ACKNOWLEDGEMENTS

We thank Nabeel Siddiqui for his early feedback on model approach and for connecting computer science to freshwater research at Susquehanna University.

### Funding

This work was supported by the Richard King Mellon Foundation, Freshwater Research Institute, and Honors Program at Susquehanna University. The funders had no role in study design, data collection and analysis, decision to publish, or preparation of the manuscript.

### Grant Disclosures

The following grant information was disclosed by the authors:
Richard King Mellon Foundation, Freshwater Research Institute, and Honors Program at Susquehanna University.

### Competing Interests

The authors declare that they have no competing interests.

### Author Contributions

- Ian M. Rudy conceived and designed the experiments, performed the experiments, analyzed the data, prepared figures and/or tables, authored or reviewed drafts of the article, and approved the final draft.
- Matthew J. Wilson conceived and designed the experiments, performed the experiments, prepared figures and/or tables, authored or reviewed drafts of the article, and approved the final draft.

### Data Availability

The data are available at Zenodo: Rudy, I., & Wilson, M. (2024). Underwater Image/Turbidity Dataset (1.0.0) [Data set]. Zenodo. https://doi.org/10.5281/zenodo.10951021.

All associated code is available at Zenodo: Rudy, I., & Wilson, M. (2024). Turbidivision: Estimating Turbidity with Underwater Photos - GitHub Repository (1.0.2). Zenodo. https://doi.org/10.5281/zenodo.11176496.

Turbidivision app and binaries are available at Zenodo: Rudy, I., & Wilson, M. (2024). Turbidivision Web App and Binaries (1.1.0). Zenodo. https://doi.org/10.5281/zenodo.11175905.

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
