# Peer review of "Turbidivision: a machine vision application for estimating turbidity from underwater images"

_PeerJ, doi:10.7717/peerj.18254_

## Round 0.1 · original submission · Minor Revisions

As you can see, three reviewers have commented on your manuscript. It was well received, and the comments—both specific and general—mainly suggest changes to clarify the content or highlight the potential impact of the work.

Reviewer 1 ·

Basic reporting

All comments have been added in detail to the last section.

Experimental design

All comments have been added in detail to the last section.

Validity of the findings

All comments have been added in detail to the last section.

Additional comments

Review Report for PeerJ
(Turbidivision: a machine vision application for estimating turbidity from underwater images)

1. Within the scope of the study, various classification operations and regression operations were performed with deep learning using underwater images.

2. In the introduction section, the importance of the subject was mentioned very limitedly. In addition, the purpose of the study and especially the originality point should be stated more clearly.

3. The type and amount of dataset used seems sufficient within the scope of the study. However, it would be more appropriate to prefer cross-validation for a more accurate analysis of the classification operations.

4. In the model training section, it was stated that YOLOv8 was used. Although there are many different deep learning models that can be used in this context in the literature, why was this model preferred in particular? When YOLO versions are examined, it is seen that more up-to-date ones are available in the literature, why was this version preferred in particular? What are the reasons why other state-of-the-art deep learning models are not preferred in the literature within the scope of the study?

5. The use of a web application within the scope of the study is a positive situation in terms of the applicability of the study. At this point, the study proves itself in terms of usability, but there are big question marks in terms of originality.

6. The results obtained and confusion matrix outputs seem sufficient for the first stage of the study. However, attention should be paid to model diversity.

In conclusion, the study is important in terms of its subject and applicability, but attention should be paid to the sections listed above in order to clarify issues such as originality and contribution to the literature.

·

Basic reporting

In this article, the authors present a model they developed to estimate turbidity from underwater images. Their findings are robust, with an extensive set of data, good model performance, and well-explained methods. This paper's content is relevant to the field of water monitoring. However, there is substantial room for improvement in the paper's form. Many sentences are ambiguous, redundant, and do not follow scientific writing standards. Therefore, I do not recommend publishing the article as it is, but I encourage the authors to carefully revisit the form, as I believe that their findings are very promising. In this review, I used the structure provided by the PeerJ editors, and used a red/orange/green colors to evaluate each point. Each evaluation is followed by a detailed explanation.

RED: The article must be written in English and must use clear, unambiguous, technically correct text. The article must conform to professional standards of courtesy and expression.

Explanation: see comments in the document. I primarily concentrated on the abstract and the introduction to provide detailed comments, but the criticism is applicable to the whole document.

ORANGE: The article should include sufficient introduction and background to demonstrate how the work fits into the broader field of knowledge. Relevant prior literature should be appropriately referenced.

Explanation: The authors defined in the first paragraph of the introduction the relevance of turbidity in water quality assessment. However, despite using in the article statements such as line 47 “High turbidity can have a negative impact on aquatic life” or line 53 “overly turbid water is unsuitable for consumption by humans”, they did not provide any information on what is considered a high/normal/low turbidity value in the context of natural waters and drinking waters. This is a problem because the reader is not able to assess whether the measurement range (0-55NTU) is relevant in this context.

GREEN: The structure of the article should conform to an acceptable format of ‘standard sections’ (see our Instructions for Authors for our suggested format). Significant departures in structure should be made only if they significantly improve clarity or conform to a discipline-specific custom.

Comment: the article is brief, well-structured and information are presented where the reader expect them.

ORANGE: The submission should be ‘self-contained,’ should represent an appropriate ‘unit of publication’, and should include all results relevant to the hypothesis.

Explanation: The authors used the argument that historical dataset of underwater images could be analyzed with their method (line 75: “historical data from underwater images often lacks […]” or line 96 “gaining insights into historical data”). I agree that this is a very strong argument for the relevance of their model. Yet they did not provide any evidence that such data are available. I suggest citing existing dataset, and even (if possible) analyzing some of historical data with their algorithm, which would with little effort greatly improve the strength of the article.

Experimental design

The submission should clearly define the research question, which must be relevant and meaningful. GREEN: The knowledge gap being investigated should be identified, and statements should be made as to how the study contributes to filling that gap.

Comment: the introduction was well structured and highlights the importance of turbidity for water monitoring, as well as the limitations of currently available techniques for turbidity measurement, leading to the research gaps.

ORANGE: The investigation must have been conducted rigorously and to a high technical standard. The research must have been conducted in conformity with the prevailing ethical standards in the field.

Explanation:
There is, at difference instance, a lack of justification to explain why the authors choose certain methods over others:
• Line 122: “For 38% of all photos, we included the 4.2cm Secchi”. Why did you do that? What is the impact of the Secchi disk on the model performance? Did you try to train the model with images without Secchi disk? (This is relevant because for future use of the model most users will not have access to a Secchi disk).
• Line 148: “We used Ultralytics YOLOv8 classification”. Why did you selected this method? Did you try other approaches? Why do you believe that this method is suited for this task?
• Line 136: “the images were broken into 11 groups”. Why eleven? Why not 5? Did you optimite this number, or is it for practical reasons?

ORANGE: Methods should be described with sufficient information to be reproducible by another investigator.

Explanation: In general, the description about the data collection lacks information, which hinders reproducibility:
• Line 106: “whether the water was flowing”. Under which criteria was the water flowing? Did you use a flowmeter? Did you use this information in the analysis?
• What about the illumination? Did you use sunlight? Did you use the camera flash? Did you collect information about it?
• Line 115: “Images were collected by taking individual photographs […] and by saving a selection of frames from a video”. Why to acquisitions methods ? How many images are taken from each mode?

Validity of the findings

ORANGE: The data on which the conclusions are based must be provided or made available in an acceptable discipline-specific repository. The data should be robust, statistically sound, and controlled.

Explanation: I am concerned about the validity of the findings because some information about the dataset are missing. The authors used images takes from different sources, which is improving the range of application of their model. My concern is: is it possible that the model is not actually predicting turbidity, but the origin of the image?
Here is a possible scenario: all samples from rivers have a turbidity between 0 and 5 NTU, all samples from the lab experiment have a turbidity between 10 and 15 NTU, etc. In that scenario, the classification model could be predicting the origin of the image, which is itself correlated with turbidity. The problem would be that if one provide a new image to the model, it will not be able to predict turbidity accurately.
Solution: 1) provide information, for each sub-parts of the dataset, about the range of turbidity. What about, in Figure X_REF, using a color-code for each dot representing the origin of the sample? 2) provide information about the train-validate-test splitting of the data, regarding whether images from each source is present in each one of them. 3) Even better: keep a whole section of the dataset, for instance river images, from the training and validating set. If the model still performs well on this totally new images, it means it learned how to estimate turbidity from any images. This is important because future users of your app need to verify that the model will still perform if they take a picture in a totally different context.

ORANGE: The conclusions should be appropriately stated, should be connected to the original question investigated, and should be limited to those supported by the results. In particular, claims of a causative relationship should be supported by a well-controlled experimental intervention. Correlation is not causation.

Comment: see comment above.

Additional comments

I have two additional remarks/questions about the result analysis.

First, why did you present the model accuracy in percentage (above 2.5 FNU)? In my opinion, the main message of this article is that with a properly trained model, any camera can give a +/- 5 FNU estimation of the turbidity, no matter the range. By giving accuracy in %, you are confusing the reader who could believe that the accuracy increases with increasing turbidity (heteroscedasticity), which is apparently not the case (I am using Figure 2, but a residual plot would enable the reader to judge heteroscedasticity more clearly).

Second, why did you analyze the results differently below and above 2.5FNU? You can consider whether this is necessary, or if it is not overcomplicating things. In my opinion, this is not very relevant, as a natural water below 2.5 FNU can already be considered as very clean, not matter if the turbidity is 1.9 or 1.8 FNU. Again, in my understanding, your approach is not aiming at being as accurate as possible, but instead to be accurate enough to provide a cheap, accessible and quick way to estimate turbidity.

Reviewer 3 ·

Basic reporting

This article presents the results of a machine vision model to estimate the turbidity from underwater images. The topic is interesting and has real world applications, specially since the authors have created a stand alone and a mobile application. The paper is well structured and clear.

Experimental design

The objective is clearly stated, which is to train a machine vision model to estimate turbidity values from underwater images.
The underwater images were taken with 2 types of cameras, which were taken in "auto" mode, hence each photo can have different exposure times and ISO. In some photos, a Secchi disk was in the field of view. Photos were collected in different water bodies: rivers, lakes, ponds and the ocean, additional photos were taken in controlled environments. A total of 675 images were taken. Reference measurements were taken with a LaMotte 2020i portable turbidity meter. To generate the model the Ultralytics YOLOv8 classification model was used.

Comments:
1. The most important part in this work is the model used, the YOLOv8, which is not explained. A brief explanation should be added.
1a. As I understand, YOLOv8 is mainly an object detection and image segmentation model, how was it applied to this research problem?
1b. It is mentioned that transfer learning was used starting with a pretrained model, which data was used for that pretrained model?
1c. It is mentioned that no changes in model accuracy where observed when the type of water body was included on the training or if a Secchi disk was in the view. Can the authors give more insights on this? All the images look very different and I wonder how the model is learning.
1d. It seems that the shutter speed and ISO do not to have any influence on the results, can the authors explain why?

2. Did the authors consider taking photos from above the water surface? Which challenges can this type of images pose?

Validity of the findings

The 675 images are provided as well as a Github repo in Zenodo (which I did not try). Additionally, stand alone and a mobile application were created, the stand alone application can be downloaded from zenodo and the mobile application from Play Store. The discussion and conclusions the results are commented and the goal of the paper is answered.

Additional comments

It is mentioned in the introduction that the approach presented herein to measure turbidity "provide an alternative to measuring turbidity when lower accuracy is acceptable, such as in citizen science and education applications". If the authors can give more information about which accuracy is needed for other users (eg. environmental agencies, municipalities) would be interesting and what would be needed to improve it.

---

## Round 0.2 · Minor Revisions

There are still some minor comments that could be easily addressed. I would particularly urge the authors to focus on the comments regarding the discussion section, as comparing approaches is especially crucial for method-oriented papers. Additionally, Reviewer 2 has clarified some of their previous remarks, providing an opportunity to reevaluate and refine the manuscript.

Reviewer 1 ·

Basic reporting

All comments have been added in detail to the last section.

Experimental design

All comments have been added in detail to the last section.

Validity of the findings

All comments have been added in detail to the last section.

Additional comments

Review Report for PeerJ
(Turbidivision: a machine vision application for estimating turbidity from underwater images)

Thank you for the revision. The new version of the paper has been reviewed in detail, with detailed responses to referee comments. Although some of the responses and changes made are very limited, when the paper is considered as a whole, it is at a significant level in terms of contribution to the literature. For this reason, I recommend that the paper be accepted. I wish the authors success in their future papers. Best regards.

·

Basic reporting

I separated this second review into three parts. First, I added in the author's response letter a detailed response on the comments that I did not considered as properly addressed. I also clarified some of my initial comments which were not sufficiently explained. Second, I added into the Word manuscript a series of comments, that, in my opinion, needs to be considered to significantly improve the manuscript. Especially the introduction can be improved by revisiting the structure, splitting paragraphs and clarifying sentences. Third, I develop here an additional point of critique is the main reason why I think that the manuscript needs further revision before being suitable for a publication.

The discussion section is not sufficient. First, I noted a lack of reference to other scientific work in the discussion section, hinting that a significant part of the discussion -putting the results in perspective with the current state of knowledge- is missing. Second, there is an important discussion paragraph missing in this paper. According to this paper, one of the main advantage of this method is that the approach enables to measure turbidity in a cheaper way (I agree with this) and also faster (this needs discussion).
As your method requires underwater images, It requires the user to attach the camera to a pole and to immerse it. In that case, I do not see how this is significantly faster that collecting a grab sample and transferring it into a portable turbidimeter. The gain of time seems quite small, as a field turbidimeter can read turbidity within seconds.
In the discussion, I would like the authors to discuss the limitations of their method, and I think that at least mentioning alternative is needed (what about using out-of-water imagery instead of underawater imagery?).

Few references to consider:

Rocher, J., Jimenez, J. M., Tomas, J., & Lloret, J. (2023). Low-Cost Turbidity Sensor to Determine Eutrophication in Water Bodies. Sensors, 23(8), Article 8. https://doi.org/10.3390/s23083913

de Camargo, E. T., Spanhol, F. A., Slongo, J. S., da Silva, M. V. R., Pazinato, J., de Lima Lobo, A. V., Coutinho, F. R., Pfrimer, F. W. D., Lindino, C. A., Oyamada, M. S., & Martins, L. D. (2023). Low-Cost Water Quality Sensors for IoT: A Systematic Review. Sensors (Basel, Switzerland), 23(9), 4424. https://doi.org/10.3390/s23094424

Li, Y., Wang, X., Zhao, Z., Han, S., & Liu, Z. (2020). Lagoon water quality monitoring based on digital image analysis and machine learning estimators. Water Research, 172. https://doi.org/10.1016/j.watres.2020.115471

Russell, S. L., Marshallsay, D. R., MacCraith, B., & Devisscher, M. (2003). Non-contact measurement of wastewater polluting load—The Loadmon project. Water Science and Technology, 47(2), 79–86. https://doi.org/10.2166/wst.2003.0090

Bersinger, T., Le Hécho, I., Bareille, G., Pigot, T., & Lecomte, A. (2015). Continuous monitoring of turbidity and conductivity in wastewater networks: An easy tool to assess the pollution load discharged into receiving water. Revue Des Sciences de l’Eau, 28(1), 9–17. Scopus. https://doi.org/10.7202/1030002ar

Experimental design

Nothing to add, few comments on the manuscript.

Validity of the findings

The findings needs to be discussed alongside other recent developments on turbidity measurement in waters. See above.

Additional comments

None

---

## Round 0.3 · accepted · Accept

Congratulations!

I think that the manuscript is ready for publication